# Sparsity Programming: Automated Sparsity-Aware Optimizations in Differentiable Programming

**Shashi Gowda**
Massachusetts Institute of Technology

**Valentin Churavy**
Massachusetts Institute of Technology

**Alan Edelman**
Massachusetts Institute of Technology

**Yingbo Ma**
Julia Computing

**Christopher Rackauckas**
Massachusetts Institute of Technology

## Abstract

Previous studies in numerical analysis have shown how the calculation of a Jacobians, Hessians, and their factorizations can be accelerated when their sparsity pattern is known. However, accurate Jacobian and Hessian sparsity patterns cannot be computed numerically, leaving the burden on the user to provide them. In this manuscript we develop a method for the accurate and efficient construction of sparsity patterns by transforming an input program into one that computes the sparsity pattern of its Jacobian or Hessian. Our implementation, which we demonstrate on partial differential equations, is a scalable technique for acceleration of automatic differentiation on arbitrarily complex multivariate programs. This work also demonstrates that the effectiveness of dynamic program analysis when applied to differentiable programming is yet to be fully realized.

## 1   Introduction

Scientific and engineering models are no longer simple mathematical functions but instead comprise of intricate programs with interconnected modules. Because of their programmatic structure, it may be nearly impossible to directly write down their mathematical form. In addition, given the high degree of collaboration and interdisciplinary work, it may be difficult for any one author to know the entirety of a model, as is demonstrated in many modular climate models like CCSM3 [7]. However, it is becoming increasingly common to want to solve inverse problems for complex programmatic models which may contain concepts from many disciplines including differential equations and neural networks [3, 15, 1]. Thus automated program analysis must replace manual mathematical analysis in order to derive properties such as Jacobians and adjoints of the model.

In this manuscript, we introduce sparsity programming: a method for automatic detection of Jacobian and Hessian sparsity in high level codes that are to be automatically differentiated. By propagating metadata through a program execution in a just-in-time compiled language, our methodology is able to efficiently build an accurate sparsity pattern.

Differentiable programming has seen the effectiveness of dynamic program analysis in deriving gradients of programs without restricting the programming language [14]. However, this may just be the tip of the iceberg of what program analysis techniques can enable in this area. Our work showcases that it is possible to utilize dynamic program analysis to perform a new set of mathematical analysis on differentiable programs.

Preprint. Under review.

## 1.1 Previous work and our contribution

Traditional numerical analysis has documented how knowledge of the sparsity in the Jacobian or Hessian of a nonlinear function $f$ can be used to compute them with lesser effort. They use ideas like matrix coloring to reduce the total number of operations required to compute Jacobians/Hessians of known sparsity [9, 6]. This previous work focused on application domains where $f$ had a strong mathematical interpretation, such as a known PDE with a tangible mathematical description, and practitioners exploited the structure of PDE discretizations to analytically know the sparsity pattern of their program. However, recently the applications of automatic differentiation has given rise to differentiable programming, a domain where derivatives are calculated through arbitrary numerical programs in dynamic languages using forward or reverse mode automatic differentiation. In such models, the sparsity pattern may be too difficult to derive by hand, but also not amenable to static sparsity analysis of traditional approaches like ADOL-C [19, 10] and TAF [11].

A pure-numerical algorithm will not suffice to find the sparsity pattern for an arbitrary program. Some reasons for this are:

- If a program has state-dependent branching, any one input could extract the sparsity pattern for the program path it happens to execute but not of the entire program.
- Extraneous zeros could be computed simply due to floating point error.
- True zeros may be hard to detect in larger programs due to floating point round-off, and thus a non-zero tolerance will need to be used to threshold values, possibly introducing false zeros which grow the sparsity pattern.

Our approach avoids these problems by regarding sparsity detection as dynamic program analysis problems. We use non-standard interpretation, the act of altering the semantics of a program's variables and operators to compute alternative results, to detect the sparsity without numerical error. In Section 2.1 we show how the Jacobian sparsity pattern can be estimated by tracking which components of the input to $f$ "influenced" which of its output components. In Section 2.2 we augmented the technique with linearity information to find the Hessian sparsity. In Section 3, we detail a procedure for correctly handling code with branching. In Section 4 we demonstrate a real-world example of our Jacobian sparsity detector. We conclude in Section 5 by discussing how this methodology is being integrated with automatic differentiation to achieve automated program acceleration in inverse problems.

## 2 Automatic Sparsity Detection through Non-Standard Interpretation

Our non-standard interpreter works by attaching metadata, called a "tag", to values produced in the program when necessary. We use the notation $\texttt{x}\langle\texttt{tag}\rangle$ to denote a tagged value: x is the value part seen by the standard interpretation of the input program, while the part inside the $\langle\rangle$ brackets is the tag part, and is only visible to our non-standard interpreter which sits on top of the standard interpreter.

### 2.1 Jacobian sparsity detection pass

We compute Jacobian sparsity by tracking which input components have a role in producing which output components. If an output j is influenced by input i, then we note that the Jacobian *could be* non-zero at the index $(i, j)$. Any $(i, j)$ pairs not discovered to be non-zeros in this way are guaranteed to be actually zeros. We track this information by tagging values in the program with a "provenance set" when necessary. This set contains the indices of the input components that influenced the value it is tagging.

Our interpreter takes a function $\texttt{f(Y, X, params...)}$ where, by convention, Y is the output vector which is mutated by the program, the X is the input vector, and $\texttt{params}$ are arbitrary parameters. The interpreter also must be given the arguments to "seed" the analysis, we use the arguments for the shape and type information they provide, our sparsity computation itself assumes that their values are subject to change.

The interpreter first tags Y and X with the tags $\texttt{Output()}$ and $\texttt{Input()}$ respectively. $\texttt{params}$ are tagged with an empty provenance set – this signifies that the parameters are subject to change, but they aren't from any input component. Then following program transforms are applied:

- **Input array access:** $X\langle\texttt{Input()}\rangle\texttt{[i]} \rightarrow \texttt{X[i]}\langle\{\texttt{i}\}\rangle$
- **Function call:** $\texttt{g(a}\langle\texttt{a\_set}\rangle\texttt{, b}\langle\texttt{b\_set}\rangle\texttt{, } \cdots\texttt{)} \rightarrow \texttt{g(a,b, } \cdots\texttt{)}\langle\texttt{a\_set}\cup\texttt{b\_set}\cup\cdots\rangle$
  For function calls, if no arguments are tagged, the rewrite is not triggered; and if not all arguments are tagged, we treat the other arguments as having an empty provenance set.
- **Output array write:** $\texttt{Y}\langle\texttt{Output()}\rangle\texttt{[j]} = \texttt{a}\langle\texttt{a\_set}\rangle \rightarrow$

```
for i in a_set
    execution_context.sparsity[i,j]=1
end
Y[j] = a
```

Here `execution_context` is a metadata object that is specific to the nonstandard execution of the function $f$. After the execution `execution_context.sparsity` object, which starts off as all zeros, contains the sparsity pattern for the program.

## 2.2 Hessian sparsity detection pass

In similar vein, we extract Hessian sparsity of a $\mathbb{C}^n \rightarrow \mathbb{C}$ function by another non-standard interpreter. This interpreter takes a function `f(X, params...)` where, by convention, X is the input vector, `params` are changeable parameters. In the hessian case, instead of tracking a provenance set, we track a *provenance polynomial* which has the same hessian sparsity pattern as the value that is tagged. The interpreter first tags X as $X\langle\texttt{Input()}\rangle$, and `params` with the polynomial that denotes the number 1 to consider them subject to change. The program is transformed as below and executed:

- **Input array access:** $X\langle\texttt{Input()}\rangle\texttt{[i]} \rightarrow \texttt{X[i]}\langle\texttt{i}^1\rangle$
- **1-argument linear function call:**
  $\texttt{g(a}\langle\texttt{a\_poly}\rangle\texttt{)} \rightarrow \texttt{g(a)}\langle\texttt{isnzderiv(g, (1,1)) ? a\_poly : a\_poly}^2\rangle$
- **2-argument linear function call:** $\texttt{g(a}\langle\texttt{a\_poly}\rangle\texttt{, b}\langle\texttt{b\_poly}\rangle\texttt{)} \rightarrow$

```
begin
    a_contribution = isnzderiv(g, (1,1)) ? a_poly :  a_poly²
    b_contribution = isnzderiv(g, (2,2)) ? b_poly :  b_poly²
    prod = isnzderiv(g, (1,2)) ?  0 :  a_poly * b_poly
    g(a, b)⟨a_contribution + b_contribution + prod⟩
end
```

Here we limit rewrites to the functions g which are known mathematical functions. We expect every other function call to call such a function somewhere down its call stack. `isnzderiv(g, (i,j))` tells us if $\partial_{ij}g$ is non-zero. Table 1 shows example hessians on very simple program fragments operating on an input vector of length 4. We store the provenance polynomial as a set of dictionaries where each dictionary is a term of the polynomial, and it containing 42=>2 says that `x[42]^2` is a factor of that term. We also canonicalize them to make sure they do not grow redundantly. We extract the sparsity of the hessian by computing the sparsity of the provenance polynomial that the result is tagged with.

## 3 Dealing with control flow

If a branch condition ends up being tagged with a provenance set or a provenance polynomial, then we know that it is subject to change in a different execution of the function. Our interpreter first runs the program setting the condition to true, and then again setting the condition to false. Since we are only interested in the sparsity pattern and not the standard result of the program, this technique is usually able to infer the sparsity of the entire program. Although this works for most real-world applications we have tried, we note the following limitations:

- There's a combinatorial increase in the number of analysis passes as the number of tagged branch conditions increases. In our experience, however, it is hard to find programs that contain more than a couple of branches that get tainted.

- `s = 0; while s < 1; s += X⟨Input()⟩[i]; end`
  This program, with a loop condition that depends on the input, is a pathological one. Here, our interpreter can go into an infinite loop trying to guess true and false at every iteration but never being able to terminate the loop. However, common use cases which may require sparsity handling, like differential equation and machine learning models, do not tend to have value-dependent control flow and thus this has not been a problem in our use cases.

- Some program paths that are forced to be taken by the interpreter may crash due to assumptions about the value within a branch. For example, in
  `foo(y, x, t) = if t < 0; y[2] = x[1] + sqrt(-t); end`
  the interpreter will override the branch `t<0` to be true and when seeded with `t=1`, `sqrt(-1)` will cause a domain error. We are working on a concolic execution [5, 4, 12, 18] approach, that should allow us to obtain appropriate program inputs (in the example, a negative value of `t`) that can then be used to legitimately explore different program paths. Since concolic execution works by querying a SAT solver for feasible values, this approach will, as a side effect, prune the space of all possible branch combinations to only those that need to be explored.

## 4   Computational Experiment: The Brussulator Semilinear Partial Differential Equation

Here we demonstrate the effectiveness of this approach on an existing chemical kinetics model applicable to climate modeling and systems biology. The Brussulator is a two dimensional ($N \times N$) nonlinear parabolic partial differential equation which describes the spatial evolution of a chemical reaction system. The system of equations is:

$$\frac{\partial u}{\partial t} = p_2 + u^2 v - (p_1 + 1)u + p_3(\frac{\partial^2 u}{\partial x^2} + \frac{\partial^2 u}{\partial y^2}) + f(x, y, t),$$

$$\frac{\partial v}{\partial t} = p_1 u - u^2 v + p_4(\frac{\partial^2 u}{\partial x^2} + \frac{\partial^2 u}{\partial y^2}),  \tag{1}$$

where

$$f(x, y, t) = \begin{cases} 5 & \text{if } (x - 0.3)^2 + (y - 0.6)^2 \leq 0.1^2 \text{ and } t \geq 1.1 \\ 0 & \text{else,} \end{cases}  \tag{2}$$

with no-flux boundary conditions and $u(0, x, y) = 22(y(1-y))^{3/2}$ with $v(0, x, y) = 27(x(1-x))^{3/2}$ [13]. This PDE is discretized to a set of $N \times N \times 2$ ODEs using the finite difference method. Listing 1 showcases the time evolution code for use with DifferentialEquations.jl [16]. It utilizes higher order code generation techniques like Cartesians, mutation, and complex control flow in order to be very efficient. Figure 1 shows the sparsity pattern as a sparse diagonal matrix corresponding to the known sparsity pattern of the 5-point stencil of the Laplacian for a system of PDEs. This showcases that our methodology is able to directly analyze the sparsity pattern of performance optimized codes, directly bringing sparsity programming to the user instead of requiring specific programming patterns.

```
1  brusselator_f(x, y, t) = ifelse((((x-0.3)^2 + (y-0.6)^2) <= 0.1^2) && (t >= 1.1), 5., 0.)
2  limit(a, N) = a == N+1 ? 1 : a == 0 ? N : a
3  function brusselator_2d_loop(du, u, p, t)
4    A, B, alpha, xyd, dx, N = p; alpha = alpha/dx^2
5    @inbounds for I in CartesianIndices((N, N))
6        i, j = Tuple(I)
7        x, y = xyd[I[1]], xyd[I[2]]
8        ip1, im1, jp1, jm1 = limit(i+1, N), limit(i-1, N), limit(j+1, N), limit(j-1, N)
9        du[i,j,1] = alpha*(u[im1,j,1] + u[ip1,j,1] + u[i,jp1,1] + u[i,jm1,1] - 4u[i,j,1]) +
10                   B + u[i,j,1]^2*u[i,j,2] - (A + 1)*u[i,j,1] + brusselator_f(x, y, t)
11        du[i,j,2] = alpha*(u[im1,j,2] + u[ip1,j,2] + u[i,jp1,2] + u[i,jm1,2] - 4u[i,j,2]) +
12                   A*u[i,j,1] - u[i,j,1]^2*u[i,j,2]
13    end
14  end
```

Listing 1: DifferentialEquations.jl-compatible time evolution code for Equation 1

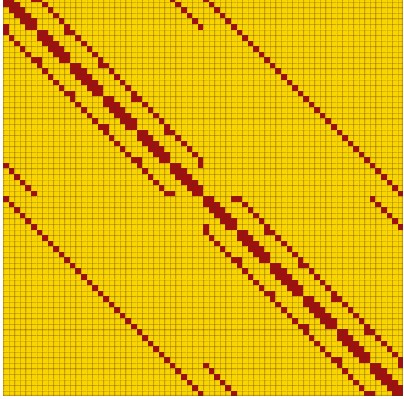

Figure 1: Sparsity pattern of the Jacobian of the Brusselator code in Listing 1 with input and output tensors of size $6 \times 6 \times 2 = 72$.

Table 1: Hessian sparsity construction for a program taking as input a vector of length 4. The $4 \times 4$ sparsity pattern for each intermediate value is shown. The provenance polynomial has the same hessian sparsity pattern.

| code fragment | polynomial | sparsity |
| --- | --- | --- |
| `deg2rad(x[1])` | $x_1$ | |
| `log(x[1])` | $x_1^2$ | |
| `x[1] + x[4]` | $x_1 + x_4$ | |
| `x[1] * x[4]` | $x_1 x_4$ | |
| `q = x[1]/x[4]` | $x_1 x_4^2$ | |
| `asin(q)*x[3]` | $(x_1^2 x_4^2)x_3$ | |

## 5  Integration with Automatic Differentiation and Beyond

Requiring users of scientific programs to know the sparsity pattern in order to get efficient results is not a method which will scale with the growing complexity of scientific simulations. Here we develop a rigorous method which is able to accurately and efficiently determine the sparsity pattern of the Jacobian and Hessian for an arbitrary Julia program. While it does have the limitation that the control flow cannot be state dependent (such as a while loop which requires knowing the value in order to know the number of iterations), the method is able to be rigorous on a large number of existing pure-Julia codes in the Julia ecosystem, thus augmenting differentiable programming with the ability to specialize on sparsity patterns.

Integration with automatic differentiation (AD) is provided by the SparseDiffTools.jl[1] package which performs matrix coloring to automatically deduce sparsity-efficient methods for calculating Jacobians and Hessians. Additionally, these patterns can be automatically analyzed to be convert general sparsity patterns to special matrix types, allowing matrix multiplication and factorization to automatically make use of specialized methods defined in packages like BandedMatrices.jl[2] and BlockBandedMatrices.jl[3] for even more efficient calculation. Packages like DifferentialEquations.jl [16] allow these patterns to be given and combine all of these operations within a higher level mathematical operation to allow for user-friendly but efficient modeling.

Advances in these techniques are leading to discussions about whether DSL-based APIs, such as that seen with mathematical optimization package JuMP [8], can instead be fully and optimally implemented without a DSL by utilizing similar dynamic program analysis. For example, a similar context as the Hessian sparsity could be used to deduce whether a program is quadratic or convex, allowing more advanced optimization tools to be automatically applied without user intervention or the constraint of a DSL. Initial results are promising.

## Acknowledgments

The modular and composable ecosystem of Julia only has power due to the countless individuals who have dedicated their time towards making Julia packages, and we thank every contributor to this wonderful community. We would like to especially thank Jarrett Revels for his work on Cassette.jl [17], and help with using it. It would be much harder to envision this project without the uplifting existence of Cassette.jl. Additionally, we would like to thank the Julia organization and Jesse Perla for the Julia Seasons of Contributions which helped fund the sparsity integration with automatic differentiation.

---

[1]https://github.com/JuliaDiffEq/SparseDiffTools.jl

[2]https://github.com/JuliaMatrices/BandedMatrices.jl

[3]https://github.com/JuliaMatrices/BlockBandedMatrices.jl

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

# 6 Appendix: implementation notes

Here we briefly describe functions and function application in the Julia language.

Every function `f` has the type `typeof(f)`, and may have one or more implementations called "methods" which are specific to the types of the arguments it is called with. The implementation (also called the body of the method) is a sequence of statements like function call, assignment, or control flow statements (`if` conditions and `while` loops). When a function is called, informally speaking, the method that is the most specific to the given argument types is chosen to be run. For full details of the method selection heuristics, refer to [2], but the "most specific" rule of thumb should be enough to understand the contents of this paper. We use the notation `f(a::Int, b::String, c::Any)` to refer to, for example, a method of the function `f` whose body is executed when called with an integer `a`, a string `b` and an object `c` of any type.

To execute a method, Julia turns the body of the method into an intermediate representation (IR) which is then transformed into into machine code. A key feature of Julia that makes our work possible is that this IR can be accessed and manipulated by the programmer to implement non-standard semantics to existing programs. We call such a transformation a dynamic compiler pass.

## 6.1 Cassette.jl: How to watch a program run

Our implementation is greatly simplified by Cassette.jl [17], a utility for writing dynamic compiler passes through contextual dispatch.

A basic Cassette compiler pass rewrites every function call `g(x...)` as `overdub(ctx, g, x...)`, which by default, in turn calls g(x...). But one can define a specialized method, for instance, `overdub(ctx::MyContext, ::typeof(+), x...)` to change the behavior of the program from its default behavior when specifically the + fucntion is called. This is called "overdubbing". `ctx` is a context object whose type is specific to the compiler pass being applied, and it can store arbitrary metadata (for example, `execution_context.sparsity` used in Section 2.1) required by the pass. Also, the context's type aggregates `overdub` methods pertaining to a specific use case, and allows multiple pass to be recursively applied. For example, we have the `JacobianSparsityContext` to define the non-standard interpretation context for Jacobian sparsity detection. This context object is initially passed to the function whose Jacobian sparsity is to be found, and Cassette takes care of passing the same object to every `overdub` method called.

Cassette also supports tagging values in the program with arbitrary metadata. These tags are transparent to the standard Julia interpretation of the program, even as values may go in and out of structs, but can be accessed or set in `overdub` methods for the purposes of our compiler pass.

## 6.2 Implementation non-standard interpreters with Cassette.jl

The the above two features of Cassette lets us (1) intercept function calls, (2) tag values with metadata. The program transformation rules in section 2 map one-to-one to `overdub` method definitions.

Due to Julia's dynamic multiple-dispatch, the data-flow-driven type inference, efficient machine code is generated on the transformed code in a just-in-time fashion. Julia hence excels in being an ideal versatile language for language hacks such as those demonstrated in this project.

