# OpenReview forum: "Sparsity Programming: Automated Sparsity-Aware Optimizations in Differentiable Programming"
_NeurIPS.cc/2019/Workshop/Program_Transformations — Program Transformations @NeurIPS2019 Poster_

### Official Review · AnonReviewer1 · 2019-09-27
**Straightforward use of operator overloading (non-standard interpretation) for sparsity detection**

**Confidence:** 4
**Rating:** 6

**Review:**

The authors implement an operator overloading method in Julia which detects sparsity. Variables are tagged with their dependencies and these are propagated throughout the program. For Hessians, the framework requires a set of mathematical primitives which are marked as linear or non-linear. A few things that are not discussed: How are opaque functions dealt with (e.g., if BLAS's AXPY primitive is used to add two vectors, the interpreter will not be able to detect individual array accesses)? And g(a[a_set], b[b_set]) -> g(a, b)[a_set | b_set] seems to make an assumption that the program is purely functional (e.g., what if g is defined as "show(b); return a;").

The paper is easy to read and the idea seems well executed. My main concern is the lack of novelty and impact. Sparsity detection is effectively finding the sources and sinks of a dependency graph, which is a common task. The implementation itself is relatively standard too (nearly identical to operator overloading in AD).

---

### Official Review · AnonReviewer2 · 2019-09-28
**Jacobian/Hessian sparsity detection via dynamic program analysis; well-written, relevant across disciplines, though novelty is unclear**

**Confidence:** 2
**Rating:** 7

**Review:**

This paper proposes a method for automatic detection of Jacobian and Hessian sparsity in complex, high-level programs through dynamic program analysis to enable more efficient automatic differentiation. The method is demonstrated on a PDE modeling chemical kinetics. I am uncertain as to the novelty of the contribution, but the paper is well-written, makes a clear case for the cross-disciplinary relevance of its contributions, contains a good exploration of pitfalls and failure modes, and fits well with the scope of the workshop.

---

### Decision · Program_Chairs · 2019-10-01

**Decision:**

Accept (Poster)

**Comment:**

The reviewers believed this was a good contribution in scope of the workshop, although it lacked real novelty.